# ICT Usage for Cross-Curricular Connections in Music and Visual Arts during Emergency Remote Teaching in Slovenia

Eda Birsa [1], Matjaž Kljun [2] and Barbara Kopačin [1,*]

1. Faculty of Education, University of Primorska, 6000 Koper, Slovenia; eda.birsa@pef.upr.si or eda.birsa@upr.si
2. Faculty of Mathematics, Natural Sciences and Information Technologies, University of Primorska, 6000 Koper, Slovenia; matjaz.kljun@upr.si
* Correspondence: barbara.kopacin@upr.si; Tel.: +386-05-6631-269

**Abstract:** Due to the COVID-19 pandemic, the entire process of teaching and learning moved online. This forced teachers and pupils to heavily rely on information and communications technology (ICT) and make adjustments to the new mode of teaching and learning in educational institutions. We conducted a qualitative case study by interviewing 24 teachers from Slovene primary schools focusing on the implementation of cross-curricular connections in music and visual arts content with the support of ICT during the period of emergency remote teaching. We found that when planning and implementing the cross-curricular learning process, teachers insufficiently took advantage of possibilities offered by modern ICT. The manner of implementing cross-curricular connections showed uncertainties in terms of understanding their specifics, resulting in the inefficient transfer of concepts taught, the results of which were seen in pupils' work. This might additionally show the negative influence of parental supervision on the creative thinking and expression of pupils. The present study emphasizes the lack of ICT competences on the part of all participants in the educational process. Our findings show the need to educate teachers by eliminating the uncertainties related to the implementation of distant cross-curricular connections while meaningfully applying ICT adapted to pupils' competences.

**Keywords:** cross-curricular integration; music art; visual art; remote teaching; ICT; interdisciplinary assignments; elementary education; learning process; music subjects; visual art subjects





## 1. Introduction

In Slovenia, as in much of the world, the entire process of teaching and learning moved online in spring 2020 due to COVID-19 pandemic. This forced teachers and pupils to heavily rely on information and communications technology (ICT) and make immediate adjustments to the new mode of teaching and learning. When incorporating ICT in the planning and implementing of the remote teaching–learning process in the first and second period of education (pupils aged six to eleven), it is necessary to think about how ICT can contribute to integrated consideration of course content, and consequently to the efficient acquisition of knowledge. However, this is not as straightforward as might be thought, as the ICT equipment in schools and that available to individuals in their homes are often very different, as are the digital competences of all involved.

In order to ensure pupils' progress in creative subjects such as music and visual arts during remote teaching and distance learning, it is even more necessary to incorporate modern teaching strategies that promote holistic learning of content with the support of suitable ICT. Modern teaching approaches, including cross-curricular connections and ICT, have already been used by teachers in the traditional implementation of the teaching–learning process. However, several ambiguities are known to exist, and inefficient connections can often be implemented in practice which do not contribute to holistic knowledge acquisition [1]. As these findings are from 2011, and focused on year five teachers as well as on

visual arts only, we wanted to expand the study by including all teachers from the first and second period of education and then to investigate whether remote teaching and increased use of ICT led to any changes in this respect. This is particularly important, as the use of modern ICT can contribute to the sustainable development of pupils' individual and technological literacy [2–4]. To this end, we conducted a study with 24 teachers who teach both music and visual arts courses to pupils in key stage 1 and 2 in order to investigate cross-curricular connections in music and visual arts content with the support of ICT during the period of emergency remote teaching.

The paper is structured as follows. The next section presents related works. Section 3 covers the materials and methods used. Section 4 presents the results, and the final section focuses on discussion.

## 2. Related Work

In the related work section, we first look at ICT in education, ICT in remote teaching, and ICT in cross-curricular connections, with a focus on Music and Visual Arts courses in the Slovenian education system. The section concludes with a discussion of the gaps in knowledge, the aims of this study, and the research questions it addresses.

### 2.1. ICT in Music and Visual Arts Education in Slovenia

ICT incorporates hardware and software that enable the functioning of video, audio, and data transfers over computer networks. All of this is possible with desktop and laptop computers, tablets, smartphones, and a plethora of applications. In the field of education, ICT can be used either in the formation of the teaching–learning process or as technical aid to deliver the content [2]. The application largely depends on the technological equipment of the environment where classes take place and the ICT competences of teachers, parents, and pupils. ICT competences determine an individual's skills regarding the use of computers and associated software, communication tools, and the internet, involving skills for incorporating technology into everyday processes, designing information, and information retrieval and evaluation [2,3,5].

Toselli [6] emphasises that ICT should never be forced, and should be gradually applied only when pupils know how to use it. Additionally, Toselli mentions synchronous (real time) technologies such as video conferencing, instant messaging, and telephone as well as asynchronous methods and forms of remote teaching and learning such as social networking services, e-mail, shared cloud documents (Google Drive https://www.google.com/drive/ (accessed on 17 November 2021) or Office 365 https://www.office.com/ (accessed on 17 November 2021)), audio and video recordings, video channels, and the web. Teachers can choose asynchronous tools to forward teaching material prepared in advance, which pupils then cover on their own, or can use synchronous video conferencing to provide live classes [7,8].

The national curriculum for music art contains recommendations for gradual and planned introduction of ICT, including television, radio, video projector, CD, MP3 and DVD players, computers, and interactive tables [9]. It advises planning the use of ICT during the preparation of the learning content and that ICT must support learning objectives while taking pupils' experiences with ICT into consideration. It states that ICT should be used as an aid for the presentation of different music-related information, while the teacher's role should not be diminished. With the transition into the second period of education it is assumed that pupils have gained experiences and understanding of elementary music theory and expression, which should be considered while embedding ICT in music activities. The advantage of using ICT is seen especially in the "transfer, storage and organization of music contents and audio records, research of sound and sound polyphony, in the transfer of acoustic performances into musical recordings, search of information and in setting up new forms of socialisation and (music) communication among users, and between users and source of (music) information" [9].

The national curriculum for visual art specifies methods of using ICT in the planning and implementation of visual art assignments. Examples include drawing in graphics editors, creating photo montages, photography, animations, or animated films, graphics in combined techniques, etc. It is up to teachers to select appropriate computer programs for pupils to efficiently and creatively take advantage of the opportunities they offer. Even software that is easy to master can support creative artistic expression and stimulate general creativity while offering the possibility of experimenting, as seen in Figure 1 [5,10]. Recently, tablet computers with stylus pens have allowed for more genuine graphic expression, while teachers can use them to motivate pupils as well as for demonstration purposes. Digital cameras are another technology that has become ubiquitous, and can be used in combination with computers to support pupils' creativity [5].

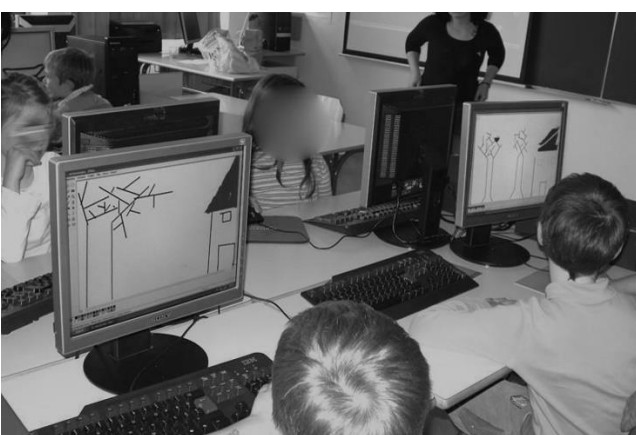

**Figure 1.** Possibilities of using ICT in the process of learning visual art in class for drawing.

*2.2. ICT and Emergency Remote Teaching in Slovenia*

Here, it is appropriate to differentiate distance learning and emergency remote teaching [11]. Distance learning is planned well in advance for when pupils and teachers are separated by time and space. Emergency remote teaching replaces traditional forms of teaching in a time of crisis, and focuses on remote delivery of content with only space being a separator. At the beginning of the COVID-19 pandemic in March 2020, emergency remote teaching became the reality, and teachers' ICT competences suddenly became very important. Both in Slovenia and elsewhere in the world, as Blahušiaková and others [12] mention, teachers and students have encountered completely new challenges. All primary school teachers and pupils in Slovenia have access to Arnes http://www.arnes.si/zavod-arnes/ (accessed on 17 November 2021) (Slovene Academic Research Network) and eAsistent https://www.easistent.com/ (accessed on 17 November 2021) services. Arnes provides the video conferencing systems Zoom https://zoom.us/ (accessed on 17 November 2021) and MS Teams https://www.microsoft.com/sl-si/microsoft-365/microsoft-teams/group-chat-software (accessed on 17 November 2021) for remote teaching as well as the Moodle Learning Management System https://moodle.org/ (accessed on 17 November 2021) (often referred to as an e-classroom) for designing and uploading teaching materials, quizzes, crosswords, and essay questions [2,13].

When planning and implementing a remote teaching and/or distance learning process, teachers must think of an efficient application of modern ICT to support various creative activities and assess what additional value ICT can provide to music and visual art subjects. For example, through creative activities, ICT can support integrated understanding of artistic content and optimal knowledge transfer involving various senses [2]. In addition, Toselli [6] argues that teachers in remote teaching have never been closer to their pupils, because they can reach each pupil via headphones or speakers, meaning that pupils have a sense of being directly addressed. As a consequence, the individualisation of learning and teaching comes even more to the forefront.

Until 2020, digital competences of pupils in the first and second period of education covered only basic skills. The use of word processors (e.g., MS Word https://www.microsoft.com/sl-si/microsoft-365/p/word/ (accessed on 17 November 2021)) and email were usually not needed in the teaching–learning process, despite the fact that these generations are called 'digital generations' [14]. The introduction of emergency remote teaching, however, required pupils and their parents/guardians to master various new ICT skills and competences [2,11,14,15].

### 2.3. ICT and Cross-Curricular Connections of Content in Music and Visual Arts

In the first and second period (key stages 1 and 2) of education in Slovenia, music and visual arts are taught by the same teacher, as the transfer of knowledge and skills can be achieved with more comprehensive consideration of concepts from both subjects using cross-curricular connections. This has been shown to have beneficial effects on the cognitive aspects of pupils [16–19]. As in individual subjects, teachers have full autonomy in the implementation of the cross-curricular teaching–learning process with music and visual art content, including choosing methods and forms of work and in planning activities with the available ICT. Cross-curricular connections need to be planned in a way that pupils are able to connect concepts already learned with new ones learned during the class [19,20].

Special attention must be paid to the planning of clear and concrete goals that can indicate the knowledge that pupils are to acquire in each particular art subject, including particular concepts of music and visual arts, concepts related to creativity, skills developed in the use of various art materials, instruments and aids, and increased aesthetic, emotional, and social qualities. Moreover, teachers must not overlook the aspect of the learning sequence, and must carefully consider the appropriateness of selected content they are planning to connect as well as appropriate learning and teaching strategies with modern ICT in order to effectively transfer knowledge at a distance. It is necessary to envisage the learning sequence of content through individual ICT, as this enables students to more easily and comprehensively understand novel musical and visual art concepts [21].

### 2.4. Purpose of the Study

Teaching methods in music and visual arts are constantly changing with offerings from novel ICT. Due to the COVID-19 pandemic in 2020, these changes were much faster than they would have been otherwise. Teachers were forced to replace even established teaching methods that did not rely on ICT with ones using ICT in order to support remote teaching and distance learning. In such a situation, cross-curricular integration represents a way of facilitating the cognitive process through the effective delivery of learning content, and can enable holistic understanding of various subjects.

When planning a cross-curricular teaching–learning process for music and visual arts, it is necessary to carefully choose the content in order to achieve the planned goals of both subjects. In traditionally-conducted teaching–learning, processes inefficient implementation of cross-curricular connections is the norm [1,22]. Irregularities and lack of knowledge of the specifics of planning and implementing connections are often issues when a musical or artistic activity is used only as a diversification of another subject instead of establishing mental strategies and knowledge transfer with equal consideration of the concepts or content of the exposed subject areas. Our goal was therefore to determine whether the 'new digital reality' encouraged teachers to consider learning content holistically and how they implemented cross-curricular connections remotely with the support of ICT. In particular, we wanted to investigate which ICT teachers used for planning and implementing their lectures and which ICT they planned for pupils to use in order to complete their assignments. We wanted to uncover whether the ICT intended to be used in the implementation of the lectures and ICT in completing the assignments was indeed used as planned or whether there were discrepancies between plans and implementation where cross-curricular connections were concerned. Based on the purpose and goals of this

research, we formulated several research questions; however, in this paper we focus on the following two:

RQ1: Which ICT do teachers use, and how do they using them in planning and implementing cross-curricular connections involving music and visual arts content during remote teaching?

RQ2: Which ICT do teachers plan for pupils to use in completing their cross-curricular assignments during remote teaching?

In the following paper, we present the course and results of our qualitative case study.

## 3. Materials and Methods

To carry out the research, which took place in the 2020/2021 school year, we used the descriptive method for empirical pedagogical research. We conducted a qualitative case study, in which we included teachers from selected primary schools. We chose an approach that allowed us to gain an in-depth understanding of the subjective aspects involved in the use of ICT in learning and in teaching interdisciplinary content in music and fine arts.

### 3.1. Participants

The research involved 24 purposefully selected teachers (three men) employed in the first (from 6 to 9 years) and second (from 10 to 12 years) period of education in primary schools in the southwestern part of Slovenia. Their average age was 33.8 years ($\pm$9.08), and their average number of teaching years was 11.2 ($\pm$10.5).

### 3.2. Instrument and Procedure

For the purpose of the research, we conducted semi-structured interviews with the participating teachers in October 2020. We conducted these interviews individually through the Zoom online platform. When conducting the interviews, we used the funnel technique, starting the conversation with open-ended questions in the context of the research questions and continuing with more concrete and content-specific questions which were formed on the spot during the interview. During the conversation, we let teachers explain the strategies they used without additional sub-questions, thus avoiding any influence on the answers. Results of these interviews are quoted directly in this paper.

We reviewed the written conversations and determined the coding units based on the research objectives and the principles of grounded theory [23]. Content-related codes were connected, and defined relevant concepts from which we derived categories related to insights into how teachers used ICT in distance teaching of music and art content, namely, using ICT for planning the remote teaching–learning process, and strategies for using ICT in the implementation of cross-curricular connections in music and visual arts. Both categories were documented in the obtained teacher statements, which were left unchanged and are denoted below by italics.

Based on the presented results, we performed consensual validation with teachers [24] to confirmed that what was written was accurate and credible data and findings that reflected the actual view of the research participants.

## 4. Results

The responses were grouped into the use of ICT in (i) planning the remote teaching–learning process; (ii) implementing cross-curricular connections in teaching music and visual arts; and (iii) planning cross-curricular assignments. The following subsections cover these three groups.

### 4.1. Using ICT in Planning the Remote Teaching–Learning Process

When the teaching–learning process first moved online in March 2020, most teachers did not plan to use video conferencing tools, as they were not familiar with them. They mainly followed the now mostly outdated recommendations for the use of ICT from the previously mentioned national curricula [9,10]. Teachers commonly used word processors

and presentation programs to prepare cross-curricular materials (Figure 2), while using their own photos to explain various concepts. Several teachers prepared detailed written instructions of the teaching–learning process available to both pupils and parents. These materials were uploaded to the e-classroom (Google classrooms http://classroom.google.com (accessed on 22 November 2021), Microsoft Teams, Moodle, and eAsistent Xooltime) together with assignments and sometimes quizzes to consolidate and assess the knowledge acquired. The selection of the e-classroom platform was at first left to teachers, and it was only later that the management of individual schools decided on a unified platform.

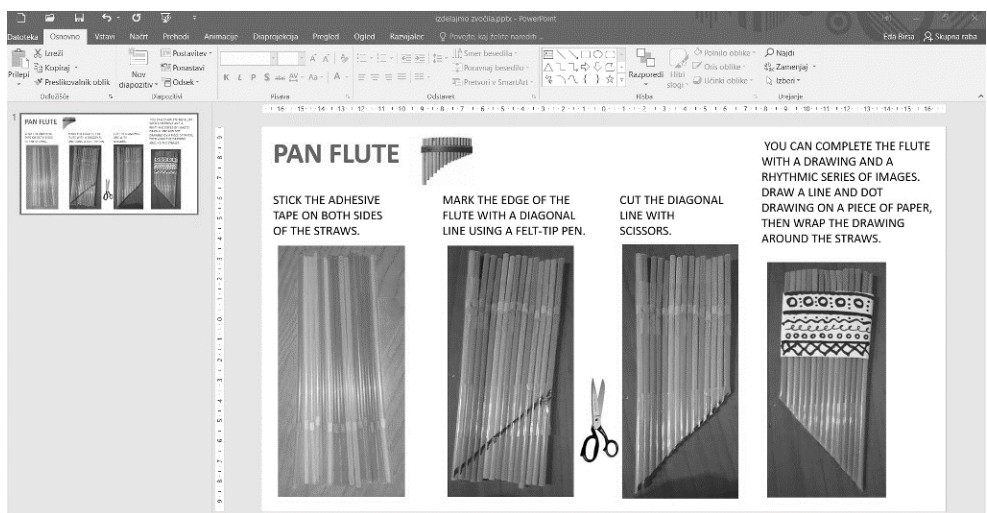

**Figure 2.** Using PowerPoint in the planning of a remote cross-curricular assignment for building a rattle, a lesson including both music and visual arts concepts.

Teachers used social networking sites and the web in general for searching and exchanging information and materials with other teachers. Multimedia materials were found on the websites of textbook publishing houses (Državna založba Slovenije (https://www.dzs.si/katalog/10237338/osnovna-sola (accessed on 2 July 2022), Mladinska knjiga https://www.ucimte.com/ (accessed on 2 July 2022) and RokusKlett https://www.rokus-klett.si/za-ucitelje/ (accessed on 2 July 2022)), on YouTube, and on teachers' own CDs. Sometimes, teachers filmed their own singing. All materials were primarily made available in the e-classroom. Teachers stressed that this preparation was based on their own knowledge of various computer programs and other ICT. Most of them pointed out a lack of ICT competences: *"... that I had too little knowledge in the field of ICT. We helped each other a lot among the teachers. I was afraid that I would not be successful in my work and that the students would not achieve the goals of the subject."* Several teachers initially needed help from family members to prepare these materials: *"I am lucky that my older children at home are much more ICT aware and willing to help when I didn't know how to use all the options that ICT offers us."*

Although offered by the Ministry of Education for decades, teachers mentioned that they had not attended ICT training for a long time, and were mainly self-educated for basic use. This is in line with a report from 2016 mentioning the lack of ICT skills among teachers in primary schools [14]. Despite the fact that various measures have been prepared at the national level to ensure and improve the digital competences of the population, especially by providing equal access to the internet and equal opportunities for all people, regardless of their abilities and acquired e-skills [2,14], a shortage and obsolescence of ICT equipment, mostly among primary school children and especially in families with many primary school children, was present in the autumn of 2020 [25].

Teachers who participated in training in the past said they had already forgotten the skills they acquired, as they did not use them in class. In addition, these training sessions did not provide the specific knowledge needed for emergency remote teaching. Additional ICT training was organised on school days after the epidemic was declared in the country.

Because grasping, understanding, and remembering everything in these short training sessions was not possible, teachers supported and helped each other, and parents came to their aid as well. With increasing familiarity teachers started to use modern ICT, which they became acquainted with through training, peer support, and the recommendation guidelines offered by [21]. Several teachers started to use ICT to identify prior knowledge and assess newly acquired knowledge; they used assignments from e-materials and e-textbooks https://eucbeniki.sio.si/ (accessed on 22 November 2021), employed services such as Socrativ https://b.socrative.com/ (accessed on 22 November 2021) and Kliker https://kliker.sio.si/ (accessed on 22 November 2021) (classroom response systems), and used more advanced functions of the available software, such as the 'immersion reader' in Microsoft Teams, which helped year 1 and year 2 pupils to listen to written instructions (Figure 3, allowing them to work more independently and flexibly [26,27].

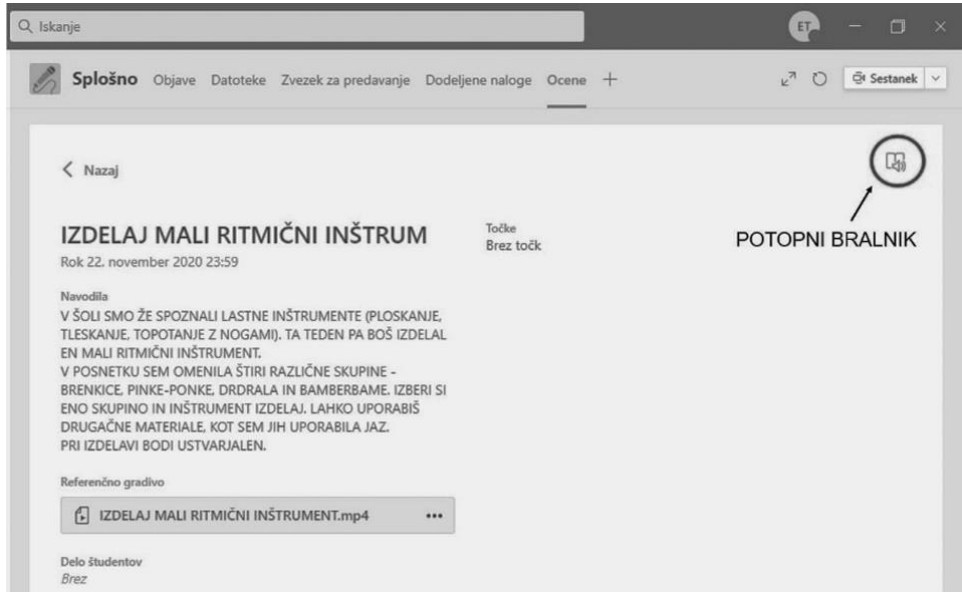

**Figure 3.** Use of Microsoft Teams e-classroom in the planning of a remote cross-curricular assignment including music and visual art.

When presenting and mentioning the possible usage of audio and video editing software (e.g., Audacity https://www.audacityteam.org/ (accessed on 22 November 2021), OpenShot https://www.openshot.org/ (accessed on 22 November 2021)), mind or concept maps (e.g., SimpleMind https://simplemind.eu/ (accessed on 22 November 2021), Creately https://creately.com/ (accessed on 22 November 2021), Coggle https://coggle.it/ (accessed on 22 November 2021), e-listovnik https://listovnik.sio.si/ (accessed on 22 November 2021)), music composition and notation software (e.g., MuseScore https://musescore.org/ (accessed on 22 November 2021) seen in Figure 4), graphics editors (Microsoft Paint, Gimp, Inkscape), and other services from the web (e.g., Artsteps https://www.artsteps.com/ (accessed on 22 November 2021) virtual galleries) as well as advanced software recommended by Breznik and Eyer [21], most participants had never heard of nor tried the majority of them. As already mentioned in the literature [2,28,29] the willingness of teachers to introduce modern approaches as well as ICT to teaching music and visual arts remains very low, despite the fact that using ICT in the process of learning can offer pupils opportunities for active and holistic learning of musical and artistic concepts [2,30].

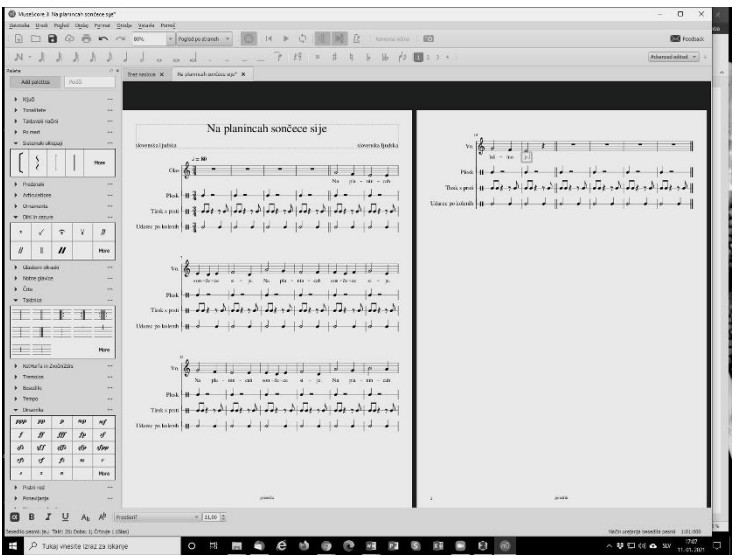

**Figure 4.** Use of MuseScore software during remote teaching.

### 4.2. Strategies for Using ICT in the Implementation of Cross-Curricular Connections in Teaching Music and Visual Arts

Teachers mostly used two ways of presenting content: (i) through the e-classroom of each particular subject by uploading text documents, presentations and audiovisual recordings, and (ii) through video conference systems (ZOOM, and to a lesser extent Cisco Webex, Microsoft Teams, Skype, and Jitsi Meet). For music art, teachers mostly presented and explained the selected musical content (concepts) via PowerPoint presentations that included audio and/or video clips (online content and local content (files and CDs) played in Windows Media Player, Microsoft Groove Music, iTunes). Several used interactive materials prepared by publishing houses. Teachers complained that they had problems with creating music as a group, as different internet connections caused delays in sound and pupils were deprived of fully experiencing sound images, that is, sound performances created with one or more instruments simultaneously [30]. In order to consolidate and test the acquired knowledge of musical contents, several teachers offered interactive quizzes. Visual art content (concepts) was mainly discussed over PowerPoint presentations. Only one of the interviewees encouraged the understanding of artistic concepts by analysing artwork found online that he included in the PowerPoint presentation.

Teachers participating in the survey mostly carried out cross-curricular connections in which music art played the leading role and visual art had a supportive role, in a similar way as presented in [4]. Our interviews revealed inadequate understanding of the leading and supporting role. For example, many teachers did not take advantage of the visual art subject for integrated transfer of knowledge because they only used it for artistic expression of the music motif, which in practice is too often understood as an appropriate implementation of cross-curricular connections; Birsa came to similar conclusions in [1].

### 4.3. Strategies for Planning and Using ICT for Pupils to Use in Completing Their Cross-Curricular Assignments

The sudden move to remote teaching in March 2020 required pupils to quickly adopt particular ICT skills, such as using e-classrooms, joining and participating in video conferences, opening, creating and editing documents, and listening to and watching audio-visual recordings, etc. on their smartphones and computers. Initially, pupils had difficulties in creating the planned assignments due to inadequate or non-existent ICT, however, these were gradually eliminated. Teachers and schools then had the possibility to better prepare for remote teaching for the school year starting in September 2020 [25].

Teachers described their pupils as often sending in recordings of assignments such as singing and playing small rhythmic instruments (Figures 5 and 6) via e-mail, the web

file sharing service we-transfer (https://wetransfer.com/) (accessed on 22 November 2021), the document sharing system Google Drive, the instant messaging system Viber https://www.viber.com/ (accessed on 22 November 2021), and Microsoft Teams. Teachers mentioned that pupils wanted to show their products, both music and visual artworks, live via video conference in order to obtain live feedback from teachers. An important observation from teachers was that the artistic motifs in these recordings were not depicted in accordance with the artistic development of pupils, which indicates the influence of parents on the implementation of artistic assignments. Their works revealed uncreative thinking and the use of art techniques in pupils' works that are not typical of children in the first and second period of education. Several teachers said: "*Because I know the students and their abilities I was surprised that some interdisciplinary assignments were unexpectedly and surprisingly accomplished.*"

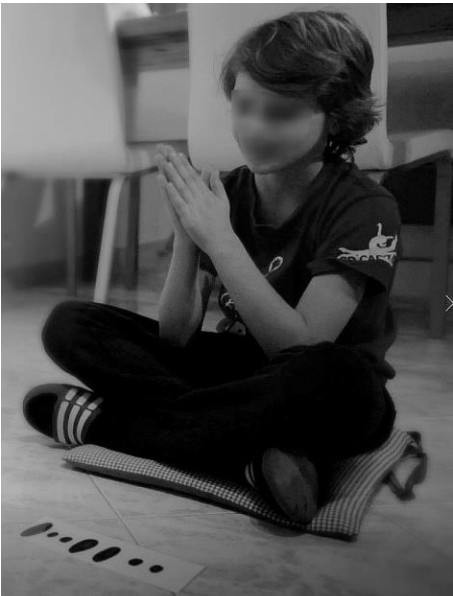

**Figure 5.** A clip from a video of a ten-year-old pupil performing a rhythm task.

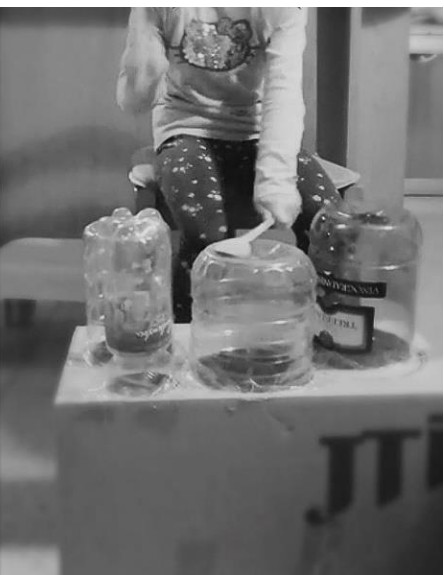

**Figure 6.** A clip from a video of a six-year-old pupil performing a rhythm task on her home-made rhythmic instrument.

Several teachers converted their lecture presentations to videos and uploaded them to YouTube in order to enable convenient viewing, which was welcomed by both parents and pupils. While pre-recorded explanations recommended by Kustec et al. [31] allow pupils to watch the content multiple times, it has been argued that this method is not particularly suitable for music and visual arts pupils in the first and second period of education due to the specifics of the subjects [9,10]. The teaching–learning process of both art subjects emphasises teacher–pupil interaction, and one-way communication fully supports a process in which pupils are focused on creative thinking and making as well as on cross-curricular integration [19].

This was observed in pupils' creations following cross-curricular assignments. Figure 7 shows an example of inefficient cross-curricular connection implemented with respect to the concept of rhythm being taught during a particular lecture. Pupils did not have the opportunity to learn the concept holistically through a pre-recorded video. While the goals of the music art lecture were achieved, the visual artwork served mainly for decorating the instrument. The emphasis of the latter was on depicting the motif with stereotypical images rather than on becoming familiar with and connecting a new visual art concept, characterised by different but constantly repeated rhythmic sequences. Teachers made insufficient use of ICT to support a holistic consideration of the concept being taught and to provide a variety of options for visual expression with computer tools to create different rhythmic sequences (multiplying images and shapes, changing colours, mirroring) using software such as MS Paint.

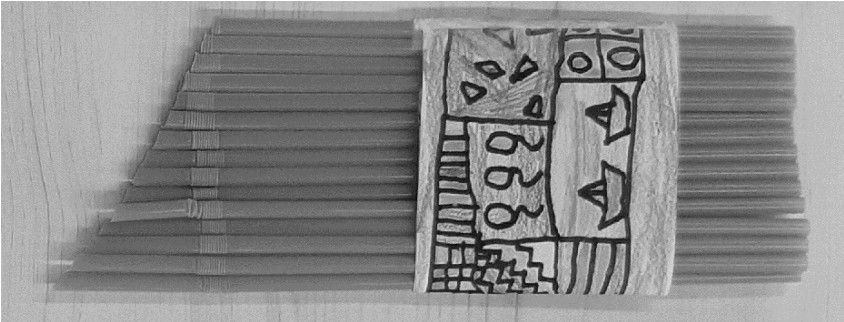

**Figure 7.** 'Pan's flute' created by a year one pupil, with rhythmically distributed lines and points.

## 5. Discussion

The following discussion section is divided into subsections based on the two research questions presented in Section 2 and the summary of the results presented in Figure 8. The third subsection highlights limitations and future directions.

### 5.1. Which ICT Do Teachers Use, and How Do They Use Them in Planning and Implementing Cross-Curricular Connections of Music and Visual Arts Content during Remote Teaching?

The two main problems that we identified in our study are lack of knowledge about how ICT can be used in the teaching and learning process, and understanding of cross-curricular connections between music and visual arts during remote teaching.

Despite the increased use of computers, the internet, and various computer software programs in education, this does not mean that they are utilised effectively or that teachers use them to their full potential. The lack of in-depth ICT knowledge is evident from the teachers' answers, which show that the ICT skills of most respondents did not exceed the use of basic computer software. According to teachers' statements about the diversity of choice in computer tools and programs, they lacked knowledge of the possibilities that modern ICT offers for creating efficient cross-curricular connections, which is in line with other studies [2,3,10,32]. Bohak Adam and Metljak [33] investigated Slovenian teachers' competences and use of ICT in the field of music during the COVID-19 pandemic. They investigated whether teachers had any problems in transferring to online music teaching and how or if they had improved their knowledge of ICT. They found no statistically

significant differences between teachers with different professional experience with respect to having problems with ICT. In addition, they showed that teachers had technical problems and problems with communication with students and emphasised other problems, such as students' unresponsiveness and ignorance or lack of knowledge concerning ICT.

| ICT in Planning the Remote Teaching-learning Process | ICT in Implementation of Cross-curricular Connections in Teaching Music and Visual Art | ICT for pupils to use in completing their cross-curricular assignments during remote teaching |
|---|---|---|
| • video conferencing tools were not used at first due to poor knowledge<br>• individual choice of tools used before national guidelines were given<br>• following outdated recommendations for the use of ICT at the beginning of remote teaching<br>• self-education of ICT for the basic use prevailed<br>• obvious shortage and obsolescence of ICT equipment<br>• forgotten ICT skills acquired in the past<br>• outdated ICT skills acquired in the past<br>• some usage of modern ICT when schools organised ICT education for teachers<br>• different understanding and adherence to the recommendations of the guidelines<br>• the use of ICT depended on the competencies of the individual<br>• help from family members often required | • content presented in synchronous and asynchronous manners<br>• teachers used interactive materials<br>• technology presented problems during group music<br>• inadequate understanding of the leading and supporting role of subjects in cross-curricular connection with or without the use of ICT<br>• inappropriate implementation of cross-curricular connections with or without the use of ICT | • difficulties in creating the planned assignments due to inadequate, obsolete or non-existent ICT at the beginning of remote teaching<br>• the assignments were never presented to teachers or the class but only submitted to a platform of choice<br>• lectures and instructions for assignments were mostly given over pre-filmed videos with no possibility to immediately ask additional questions<br>• pre-filmed videos gave a possibility to cover the materials several times<br>• students showed interest for live feedback from teachers<br>• the obvious influence of parents on implementation of artistic assignments was shown<br>• flexible completion of assignments independent of when the course was on the timetable |

**Figure 8.** The main findings summarised and divided based on the research questions.

Despite the fact that the curricula of both art subjects from 2011 and the guidelines for the use of ICT which were provided to teachers in 2020 contain recommendations for ICT usage, the interviewees mostly did not follow them. The interviews revealed that teachers did not always take advantage of all the possibilities of cross-curricular connections available with ICT to implement these connections. One said: " . . . *when we switched to distance learning, my thoughts were occupied by problems related to the use of ICT, so I probably often neglected the possibilities of interdisciplinary links between music and fine arts.*" It is true, however, that implementation of cross-curricular connections with ICT depend on both teachers' ICT competences and on the guidelines available in curricula, guidelines for the use of ICT [5,21], and other teaching materials such as interactive materials offered by publishing houses, which may or may not promote cross-curricular connections.

As already mentioned, the authors of both national curricula [9,10] and guidelines for ICT use [5,21] for music and visual arts highlight the use of ICT for knowledge enrichment, skills development, and personality formation, as well as key competences for lifelong learning. The guidelines for ICT use for music education [21] emphasize that students develop their use of digital technology, independent learning, and critical judgement. They develop digital competences in the areas of collecting, storing, sharing, and communicating information, e.g., about music, composers, etc. They learn in depth about different digital tools for online communication. The guidelines for visual art education [5] do not highlight how ICT can enrich the teaching and learning process. Rather, it mentions that pupils can develop different art skills and understanding of theoretical underpinnings with or without the help of ICT. Our study reveals that the guidelines for the use of ICT for visual arts are outdated. While the national curricula for music education include detailed cross-curricular connections, these are barely mentioned for visual arts [34]. However, neither national

curricula nor the newer national guidance for the use of ICT provide concrete information on how to implement cross-curricular lessons using ICT.

One should not overlook the fact that lack of knowledge of the content of different subject areas and of cross-curricular connection strategies can lead to inefficient and inadequate cross-curricular connections, even if teachers are highly digitally literate [14,35]. Therefore, it is very important to use the planning stage to consider strategies for transferring music and visual arts knowledge and skills in the remote educational process through meaningful use of ICT. This makes the implementation of cross-curricular connections much more efficient, as Birsa and Kopačin [29] have previously pointed out.

The experiences reported by our participants indicate that with appropriate guidelines and additional training in both ICT and cross-curricular connections, these challenges can become opportunities to learn about new themes, strategies, and ways of communicating.

### 5.2. Which ICT Do Teachers Plan for Pupils to Use in Completing Their Cross-Curricular Assignments during Remote Teaching?

An analysis of the ICT used by the pupils to complete cross-curricular assignments during distance learning showed that they used both hardware and software that they were already familiar with, as well as hardware and software that required the help of an adult. The assignments showed that teachers wanted to encourage creative expression in both music and visual arts subjects. However, especially in the visual art assignments, the problem of adults interfering with pupils' work and thus influencing their creative thinking was highlighted. Duh and Vrlič [36] emphasised that the creative learning process should encourage pupils to develop a variety of ideas in order to find new possibilities and solutions to assignments, to experiment with art techniques and materials, to enquiry various sounds on instruments, etc. This was made impossible by parents' intervention in pupils' creative cognitive processes. The fact that pupils often required help from their parents to use ICT, which is understandable considering the pupils' age, contributed to this phenomenon. Other research [37], however, has shown positive effects on the realization of the learning process when the parents of students were involved in it.

One of the reasons that parents helped students more than they would have done otherwise could be that teachers provided students with too many or too complex tasks, which they were not able to complete in the time allotted. This is precisely the problem highlighted by Toseli [6], who notes that the first reaction of teachers when adopting distance learning was to give students too many tasks, which turned out to be counterproductive and a wrong didactic decision. He says that while fewer tasks for students requires better organization and more work for the teacher, it encourages greater collaboration between teachers in different subject areas and the development of distance learning that is not just about homework and revision without the student being present. This leads to the proposal that the implementation of both synchronous and asynchronous distance learning should be oriented towards more project-based, creative, interdisciplinary, and flexible tasks. In this way, students can be more motivated and creative in the artistic field, and are able to be successful without the help of an adult.

### 5.3. Limitations and Future Work

This qualitative research is limited to participants in the Slovenian educational environment. In subsequent research, the size and the heterogeneity of the included sample in the Slovenian learning area could be increased, and perhaps teachers from other countries could be added in order to expand the research to an international context. Nevertheless, the conducted research addressed several of the practical issues that arise for teachers in connection with the implementation of interdisciplinary links between music and art content in the new digital reality. Further research could additionally examine the situation in the field of teachers' digital competences in connection with the implementation of the learning process of art subjects and the integration of new ICT skills in the return to the school environment.

## 6. Conclusions

The aim of this qualitative research was to uncover the use of ICT by teachers in the first and second period of education (key stages 1 and 2) for planning and implementing cross-curricular connections of music and visual arts content during remote teaching. Teachers initially used ICT which they had already used before emergency remote teaching began. Later, they were forced to adapt and become acquainted with the use of modern hardware, software, and online services, and had to adjust their assignments to their pupils' ICT competences. Teachers' ICT competences were not influenced by previous professional training offered nationally by the Ministry of Education, as it turned out that teachers did not attend them often, or that if they did the acquired knowledge proved to be useless for remote teaching, as it was outdated. Furthermore, teachers did not know how to take advantage of all the opportunities offered by ICT, as previously pointed out by Duh and colleagues [28].

A particular focus of this study was on using ICT in the implementation of cross-curricular connections for the remote teaching of music and visual arts. This study shows that teachers used both asynchronous (e.g., e-classroom) and synchronous (e.g., video conference) delivery of learning contents to implement the remote teaching–learning process. Teachers used interactive materials to explain novel concepts discussed in both arts subjects on certain occasions, however, they did not encourage efficient cross-curricular connections. This study revealed a lack of knowledge about the specifics of cross-curricular connections, and as such, this situation has not changed over the past ten years. Most teachers used one subject only as a motif [1], while others used one subject only as a support for the leading subject and its learning goals [36]. Cross-curricular assignments were modestly performed, and ICT was insufficiently applied.

Another objective of this study was to learn more about the planning and use of ICT for cross-curricular assignments. It turned out that parents helped their children with both ICT usage and with their assignments. Parents' influence on their child's creative thinking and expression can hinder their creative and cognitive development. This was mainly visible in pupils' final products (e.g., artwork), which did not show solutions typical of the developmental stage of children aged between six and eleven. This study emphasises the importance of teachers' live feedback on the works submitted by pupils. Students' pride in showing their creations and receiving live response is very important also in remote teaching.

In order to improve remote teaching and render cross-curricular connections effective with the application of ICT, changes and updates to the curriculum of both art subjects should be considered, at least at the primary school level. New methods and forms of work should be suggested with innovative and interactive digital learning environments, including recommendations for the reasonable application of modern ICT in the teaching–learning processes of music and visual arts. To an extent, this has already been noted in the literature [38]; however, the period of emergency remote teaching intensively stressed the importance of it. Furthermore, it is necessary to update the interactive teaching materials in the art subjects; while these are currently available on the websites of Slovenian publishing houses, they have not been updated for years. This could promote effective cross-curricular connections of both subjects [28]. In addition, it is necessary to investigate the ICT competences of pupils under the age of 16 in Slovenia, as research thus far has focused on people aged between 16 and 74 [14]. This would allow teachers to adjust the use of the appropriate ICT to pupils' age and skills, supporting the further leveraging of ICT competences.

**Author Contributions:** Conceptualization, E.B. and B.K.; methodology, E.B. and B.K.; validation, E.B., B.K. and M.K.; formal analysis, E.B. and B.K.; research, E.B., B.K. and M.K.; data curation, E.B., B.K. and M.K.; writing—original draft preparation, E.B., B.K. and M.K.; writing—review and editing, E.B., B.K. and M.K. All authors have read and agreed to the published version of the manuscript.

**Funding:** This research received no external funding.

**Institutional Review Board Statement:** The study was conducted in accordance with the Declaration of Helsinki, and approved by the Institutional Review Board of the University of Primorska (application number 2020-12 approved on 14 May 2020).

**Informed Consent Statement:** Informed consent was obtained from all subjects involved in the study.

**Data Availability Statement:** In this study we obtained qualitative data through interviews. We assured the participants that the data obtained would only be used for research purposes, not be disseminated, and be securely stored and appropriately discarded after the research is finished.

**Acknowledgments:** We wish to thank all participating primary school teachers for their cooperation in the study.

**Conflicts of Interest:** The authors declare no conflict of interest

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
