# Peer review of "ICT Usage for Cross-Curricular Connections in Music and Visual Arts during Emergency Remote Teaching in Slovenia"

_electronics, doi:10.3390/electronics11132090_

Round 1
Reviewer 1 Report
The research conducted in Slovenia emphasizes the importance and need to strengthen the cross-curricular connections of music and visual art through the improvement of the application of ICT in the first and second period of education (pupils ages 6 to 9).
The qualitatively conducted research showed more or less expected results, that teachers have underused the opportunities offered by the modern ICT. Some of the key reasons for this are analyzed, and as a result, interesting proposals are given in the concluding part, including expanding research in Slovenia and beyond. It may be interesting to compare the obtained results within different local regions (or even different countries).
However, in my opinion, it seems that not enough attention has been paid to one important fact: line 416 – 418: „Despite the fact that the curricula of both art subjects and the guidelines for the use 416 of ICT, which were given to teachers in 2020, contain recommendations for ICT usage, the interviewees did mostly not.“. Why is to so? Are the teachers and their ICT skills the only important issue? What were the guidelines and how were they delivered to the teachers? How were teachers taught to implement cross-curricular connections in practice before entering the classroom? The study uncovers some crucial problems in the emergency remote teaching usage of ICT in Slovenia, addressed by many teachers in practice, that teachers (and parents, and pupils) were forced to adapt and become acquainted with the use of modern hardware, software and online services. And, as a result their willingness to introduce modern approaches as well as ICT to teaching music and visual art is constantly very low.
Author Response
Response to review with changes
We are very grateful to the reviewers for insightful suggestions to improve the paper and several positive comments (“interesting proposals are given in the concluding part, including expanding research in Slovenia and beyond”). Below we provide the list of changes made to the text of the paper. The text copied from the paper is in italic with blue text denoting the changes. Changes are also colored blue in the paper for better understanding of what has been changed.
[Comment 1] It may be interesting to compare the obtained results within different local regions (or even different countries).
This is a good suggestion. To address it we have expanded the discussion section with the following text on page 5 Section 2.3:
Special attention must also be paid to the planning of clear and concrete goals that will indicate the knowledge that pupils will acquire in each particular art subject including particular concepts of music and visual arts, concepts related to creativity, skills developed in the use of various art materials, instruments and aids, increased aesthetic, emotional and social qualities. Moreover, teachers must not overlook the aspect of the learning sequence and must carefully consider the appropriateness of selected content they are planning to connect, as well as appropriate learning and teaching strategies with modern ICT in order to effectively transfer knowledge at a distance. It is thus necessary to envisage the learning sequence of contents through individual ICT, which will enable students to more easily and comprehensively understand novel musical and visual art concepts [21].
[Comment 2] Not enough attention has been paid to one important fact: line 416 – 418: „Despite the fact that the curricula of both art subjects and the guidelines for the use of ICT, which were given to teachers in 2020, contain recommendations for ICT usage, the interviewees did mostly not.“. Why is it so? Are the teachers and their ICT skills the only important issue? What were the guidelines and how were they delivered to the teachers? How were teachers taught to implement cross-curricular connections in practice before entering the classroom?
These are very good pointers. We have added an additional discussion section by taking some content from the results section. We have also expanded the discussion by answering the above questions. The section is too long to be included here.

Reviewer 2 Report
1. In the "Introduction", you mention "key stages 1 and 2". Please, add a sentence to explain what these stages are since you mention them for the 1st time.
2. "Introduction" seems too large for me. Judging by the "introductory" role of this section, in my opinion, it should only let the reader understand what is the topic covered in the paper, present motivation of the authors, objectives and, probably, structure. The Purpose of the Study part fits there, although may be a bit shorter. But what goes before should better fit in the section like "Literature review" or "Previous work" etc.
3. I am not sure I understand the difference between RQ1 and RQ2. ICT teachers use is anyway planned for students, isn't it? So I do not clearly understand how this ICT would differ in the 1st and 2nd options.
4. The "Results and Discussion" as one section, in my opinion, is not quite correct. Results should be presented clearly and distinguished in the paper. It is the most important part of your research. Then the discussion is a bit of a summary of what you have in the paper - of both the theoretical and the practical (survey) parts. I suggest dividing it into two separate sections.
Also in "Results", I suggest replacing so much plain text in paragraphs with tables or, at least, bullet points. Making a list of the most important ideas is a good way of visualizing your results and it catches the reader's eye much faster than searching through paragraphs.
Author Response
Response to review with changes
We are very grateful to the reviewers for insightful suggestions to improve the paper and several positive comments (“interesting proposals are given in the concluding part, including expanding research in Slovenia and beyond”). Below we provide the list of changes made to the text of the paper. The text copied from the paper is in italic with blue text denoting the changes. Changes are also coloured blue in the paper for better understanding of what has been changed.
[Comment 1] In the "Introduction", you mention "key stages 1 and 2". Please, add a sentence to explain what these stages are since you mention them for the 1st time.
This is indeed the case. Thank you for spotting this. We have tried to give a comparison to the UK system. We have removed the text and added the footnote:
1In the UK educational system these are referred to as key stage 1 (ages 5-7) and 2 (ages 7-11).
[Comment 2] "Introduction" seems too large for me. Judging by the "introductory" role of this section, in my opinion, it should only let the reader understand what is the topic covered in the paper, present motivation of the authors, objectives and, probably, structure. The Purpose of the Study part fits there, although may be a bit shorter. But what goes before should better fit in the section like "Literature review" or "Previous work" etc.
We agree with the reviewer. We have split the first section in two parts. Namely, the introduction section and related work section. We have also emphasised the purpose of the study and added the structure of the paper at the end of the Introduction section.
[...] Since these findings are from 2011 and focused on year 5 teachers as well as on visual art only, we wanted to expand the study by including all teachers from the first and second period of education and investigate whether remote teaching and increased use of ICT changed something in this respect. This is particularly important since the use of modern ICT contributes to a sustainable development of an individual and technological literacy of pupils [2–4]. To this end we have conducted a study with 24 teachers who teach both music and visual art courses pupils in key stage 1 and 2 to investigate cross-curricular connections of music and visual art contents with the support of ICT during the emergency remote teaching.
The paper is structured as follows. The next section presents related work. Section 3 covers Materials and methods. Section 4 presents the results and the last section is focusing on the discussion.
We have also added the opening paragraph to the Related work section.
In the related work section we will first look at ICT in education, ICT in remote teaching, and ICT in cross.curricular connections with the focus on Music and Visual Arts courses in the Slovenian education system. The section concludes with the gaps in knowledge and aim of the study together with research questions.
[Comment 3] I am not sure I understand the difference between RQ1 and RQ2. ICT teachers use is anyway planned for students, isn't it? So I do not clearly understand how this ICT would differ in the 1st and 2nd options.
Thank you for spotting this ambiguity. When planning for a particular lecture or the whole course different ICT tools might be used compared to tools used when implementing it (this is the tools used during teaching and the tools used by pupils to complete their assignments). The second thing is that the plans of using ICT for a particular lecture or the whole course might be too ambitious and do not get realised in their entirety as planned. Especially when cross-curricular connections are considered. We have clarified this in the text.
[...] In particular, we wanted to investigate which ICT teachers used for planning and implementing their lectures and which ICT they planned for pupils to use in order to complete their assignments. We also wanted to uncover if the ICT planned to be used in the implementation of the lectures and ICT planned to be used in completing the assignments was indeed used as planned or were there discrepancies between the plans and implementations when cross-curricular connections are considered. Based on the purpose and goals of this research, we have formulated several research questions however in this paper we focus on the following two: [...]
We have also clarified the second research question in order to make our aim more obvious:
RQ2: Which ICT did teachers plan for pupils to use in completing their cross-curricular assignments during remote teaching?
[Comment 4] The "Results and Discussion" as one section, in my opinion, is not quite correct. Results should be presented clearly and distinguished in the paper. It is the most important part of your research. Then the discussion is a bit of a summary of what you have in the paper - of both the theoretical and the practical (survey) parts. I suggest dividing it into two separate sections.
We have followed some examples of papers in the journal. However, we agree with the reviewer and have split the results and discussion section. The latter is now a separated section, which also provides more in-depth interpretations and additional works cited. The section is too long to be included here.
[Comment 5] Also in "Results", I suggest replacing so much plain text in paragraphs with tables or, at least, bullet points. Making a list of the most important ideas is a good way of visualizing your results and it catches the reader's eye much faster than searching through paragraphs.
This is a very good suggestion. We have added Figure 8 which summarises all the results from the paper, which can be seen below.

Round 2
Reviewer 2 Report
After the changes made I consider
manuscript ready for publication.